# Regularity Normalization: Constraining Implicit Space with Minimum Description Length

## Abstract

Inspired by the adaptation phenomenon of biological neuronal firing, we propose regularity normalization: a reparameterization of the activation in the neural network that take into account the statistical regularity in the implicit space. By considering the neural network optimization process as a model selection problem, the implicit space is constrained by the normalizing factor, the minimum description length of the universal code. We introduce an incremental version of computing this universal code as normalized maximum likelihood and demonstrated its flexibility to include data prior such as top-down attention. Preliminary results showed that the proposed method outperforms existing methods in tackling the limited and imbalanced data in a non-stationary setting. As an unsupervised attention mechanism given input data, this biologically plausible normalization has the potential to deal with other real-world scenarios as well as reinforcement learning setting where the rewards are sparse and non-uniform. Further studies is proposed to discover these scenarios and explore the behaviors among its variants.

## 1 Introduction

The Minimum Description Length (MDL) principle asserts that the best model given some data minimizes the combined cost of describing the model and describing the misfit between the model and data (Rissanen, 1978) with a goal to maximize regularity extraction for optimal data compression, prediction and communication (Grünwald, 2007). Most unsupervised learning algorithms can be understood using the MDL principle (Rissanen, 1989), treating the neural network as a system communicating the input to a receiver. If we consider the neural network training as the optimization process of a communication system, each input at each layers of the system can be described as a point in a low-dimensional continuous constraint space (Zemel & Hinton, 1999). If we consider the neural networks as population codes, the constraint space can be subdivided into the input-vector space, the hidden-vector space, and the *implicit space*, which represents the underlying dimensions of variability in the other two spaces, i.e., a reduced representation of the constraint space. For instance, given a image of an object, the rotated or scaled version still refers to the same object, thus each image instance of the same object can be represented by a position on a 2D implicit space with one dimension as orientaiton and the other as size (Zemel & Hinton, 1999). The relevant information about the implicit space can be constrained to ensure a minimized description length of the system.

This type of constraint can also be found in biological brains of primates: high-level brain areas are known to send top-down feedback connections to lower-level areas to select of the most relevant information in the current input given the current task (Ding et al., 2017), a process similar to the communication system. This type of modulation is performed by collecting statistical regularity in a hierarchical encoding process among brain areas. One feature of the neural coding during the hierarchical processing is the adaptation: in vision neuroscience, vertical orientation reduce their firing rates to that orientaiton after the adaptation (Blakemore & Campbell, 1969), while the cell responses to other orientations may increase (Dragoi et al., 2000). These behaviors well match the information theoretical point-of-view that the most relevant information (saliency), which depends on the statistical regularity, have higher "information", just as the firing of the neurons. The more regular the input features are, the lower it should yield the activation. We introduce the minimum description length (MDL), such that the activation of neurons can be analogous to the code length of the model (a specific neuron or neuronal population) - a shorter code length would be assigned to a more regular input (such as after adaptation), and a longer code length to a more rare input or event.

In this paper, we adopt the similar definition of implicit space as in Zemel & Hinton (1999), but extend it beyond unsupervised learning, into a generic neural network optimization problem in both supervised and unsupervised setting. Given the neuroscience inspiration described above, we consider the formulation and computation of description length differently. Instead of considering neural networks as population codes, we formulate each layer of neural networks during training a state of module selection. In our setup, the description length is computed not in the scale of the entire neural networks, but by the unit of each layer of the network. In addition, the optimization objective is not to minimize the description length, but instead, to take into account the minimum description length as part of the normalization procedure to reparameterize the activation of each neurons in each layer. The computation of the description length (or model cost as in Zemel & Hinton (1999)) aims to minimize it, while we directly compute the minimum description length in each layer not to minimize anything, but to reassign the weights based on statistical regularities. Finally, we compute the description length by an optimal universal code obtained by the batch input distribution in an online incremental fashion.

We begin our presentation in section 2, formulating the problem setting in neural network training as a layer-specific model selection process under MDL principle. We then introduce the proposed regularity normalization (RN) method, its formulation and the incremental implementation. We also present several variants of the regularity normalization by incorporating batch and layer normalizations, termed regularity batch normalization (RBN) and regularity layer normalization (RLN), as well as including the data prior as a top-down attention mechanism during the training process, termed saliency normalization (SN). In appendix A, we present the preliminary results on the imbalanced MNIST dataset and demonstrated that our approach is advantageous over existing normalization methods in different imbalanced scenarios. In the last section, we conclude our methods and point out several future work directions as the next step of this research.

## 2 PROBLEM SETTING

### 2.1 MINIMUM DESCRIPTION LENGTH

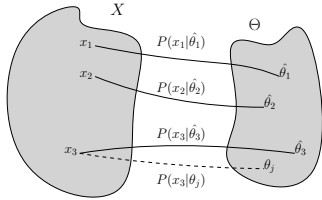

Figure 1: **Normalized maximal likelihood.** Data sample $x_i$ are drawn from the data distribution $X$ and model $\hat{\theta}_i$ is the optimal model that describes data $x_i$ with the shortest code length. $\theta_j$ is an arbitrary model that is not $\hat{\theta}_3$, so $P(x_3|\theta_j)$ is not considered when computing optimal universal code according to NML formulation.

Given a model class $\Theta$ consisting of a finite number of models parameterized by the parameter set $\theta$. Given a data sample $x$, each model in the model class describes a probability $P(x|\theta)$ with the code length computed as $-\log P(x|\theta)$. The minimum code length given any arbitrary $\theta$ would be given by $L(x|\hat{\theta}(x)) = -\log P(x|\hat{\theta}(x))$ with model $\hat{\theta}(x)$ which compresses data $x$ most efficiently and offers the maximum likelihood $P(x|\hat{\theta}(x))$ (Grünwald, 2007). However, the compressibility of the model will be unattainable for multiple inputs, as the probability distributions are different. The solution relies on a universal code, $\bar{P}(x)$ defined for a model class $\Theta$ such that for any data sample $x$, the shortest code for $x$ is always $L(x|\hat{\theta}(x))$ (Rissanen, 2001).

### 2.2 NORMALIZED MAXIMUM LIKELIHOOD

Among the universal code, the normalized maximum likelihood (NML) probability minimizes the worst-case regret with the minimax optimal solution is given by Myung et al. (2006):

$$P_{NML}(x) = \frac{P(x|\hat{\theta}(x))}{\sum_{x'} P(x'|\hat{\theta}(x'))} \tag{1}$$

where the summation is over the entire data sample space. Fig 1 describes the optimization problem of finding optimal model $P(x_i|\hat{\theta}_i)$ given data sample $x_i$ among model class $\Theta$. The models in the class, $P(x|\theta)$, are parameterized by the parameter set $\theta$. $x_i$ are data sample from data $X$. With this distribution, the regret is the same for all data sample $x$ given by Grünwald (2007):

$$COMP(\Theta) \equiv -\log P_{NML}(x) + \log P(x|\hat{\theta}(x)) = \log \sum_{x'} P(x'|\hat{\theta}(x')) \qquad (2)$$

which defines the model class complexity, i.e. how many data samples can be well explained by $\Theta$.

## 2.3 NEURAL NETWORKS AS MODEL SELECTION

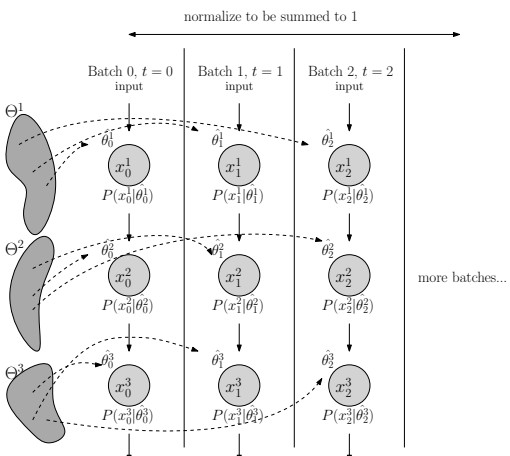

Figure 2: **Model selection in neural network.** If we consider each time step of the optimization (drawn here to be batch-dependent) as the process of choose the optimal model from model class $\Theta^i$ for $i$th layer of the neural networks, the optimized parameter $\hat{\theta}_j^i$ with subscript $j$ as time step $t = j$ and superscript $i$ as layer $i$ can be assumed to be the optimal model among all models in the model class $\Theta^i$. The normalized maximum likelihood can be computed by choosing $P(x_j^i|\hat{\theta}_j^i)$, the "optimal" model with shortest code length given data $x_j^i$, as the summing component in the normalization.

In the neural network setting where optimization process are performed in batches (as incremental data sample $x_j$ with $j$ denoting the batch $j$), the model selection process is formulated as a partially observable problem (as in Fig 2). Herein to illustrate our approach, we consider a feedforward neural network as an example, without loss of generalizability to other architecture (such as convolutional layers or recurrent modules). $x_j^i$ refers to the activation at layer $i$ at time point $j$ (batch $j$). $\theta_j^i$ is the parameters that describes $x_j^i$ (i.e. weights for layer $i - 1$) optimized after $j - 1$ steps (seen batch 0 through $j - 1$). Because one cannot exhaust the search among all possible $\theta$, we assume that the optimized parameter $\hat{\theta}_j^i$ at time step $j$ (seen batch 0 through $j-1$) is the optimal model $P(x_j^i|\hat{\theta}_j^i)$ for data sample $x_j^i$. Therefore, we generalize the optimal universal code with NML formulation as:

$$P_{NML}(x_i) = \frac{P(x_i|\hat{\theta}_i(x_i))}{\sum_{x_j}^{0,\cdots,i} P(x_j|\hat{\theta}_j(x_j))} \qquad (3)$$

where $\hat{\theta}_{[0,\cdots,i-1]}(x_i)$ (or denoted $\hat{\theta}_i(x_i)$ as in Fig 2) refers to the parameter already optimized for $i - 1$ steps and have seen sequential data sample $x_0$ through $x_{i-1}$. This distribution is updated every time a new data sample is given, and can therefore be computed incrementally.

# 3 REGULARITY NORMALIZATION

## 3.1 FORMULATION

Regularity normalization is outlined in Algorithm 1, where the input would be the activation of each neurons in certain layer and batch. Parameters $COMP$ and $\theta$ are updated after each batch, through the incrementation in the normalization and optimization in the training respectively. The incrementation step involves computing the log sum of two values, which can be easily stabilized with the log-sum-exp trick. The normalization factor is then computed as the shortest code length $L$ given the NML.

## 3.2 Variant: Saliency Normalization

NML distribution can be modified to also include a data prior function, $s(x)$, given by Zhang (2012):

$$P_{NML}(x) = \frac{s(x)P(x|\hat{\theta}(x))}{\sum_{x'} s(x')P(x'|\hat{\theta}(x'))} \tag{4}$$

where the data prior function $s(x)$ can be anything, ranging from the emphasis of certain inputs, to the cost of certain data, or even top-down attention. For instance, we can introduce the prior knowledge of the fraction of labels (say, in an imbalanced data problem where the oracle informs the model of the distribution of each label in the training phase); or in a scenario where we wish the model to focus specifically on certain features of the input, say certain texture or color (just like a convolution filter); or in the case where the regularity drifts (such as the user preferences over years): in all these applications, the procedure can be more strategic given these additional information. Thus, we formulate this additional functionality into our regularity normalization, to be saliency normalization (SN), where $P_{NML}$ is computed with the addition of a pre-specified $s(x)$.

## 3.3 Variant: Beyond Elementwise Normalization

In our current setup, the normalization is computed elementwise, considering the implicit space of the model parameters to be one-dimensional (i.e. all activations across the batch and layer are considered to be represented by the same implicit space). Instead, the definition of the implicit can be more than one-dimensional to increase the expressibility of the method, and can also be user-defined. For instance, we can perform regularity normalization over the layer dimension such that the implicit space has the dimension of the layer, as the regularity layer normalization (RLN), and similarly, if over the dimension of the batch, regularity batch normalization (RBN), which have the potential to inherit BN and LN's innate advantages.

---

**Algorithm 1: Regularity Normalization (RN)**

**Input**: Values of $x$ over a mini-batch: $\mathcal{B} = \{x_{1,\cdots,m}\}$;
**Output**: $y_i = RN(x_i)$ given **Parameter**: $COMP_t, \hat{\theta}_t$

$COMP_{t+1} = \text{increment}(COMP_t, P(x_i|\hat{\theta}_t(x_i)))$
$L_{x_i} = COMP_{t+1} - \log P(x_i|\hat{\theta}_t(x_i))$
$y_i = L_{x_i} * x_i$

---

## 4 Conclusion

Inspired by the neural code adaptation of biological brains, we propose a biologically plausible normalization method taking into account the regularity (or saliency) of the activation distribution in the implicit space, and normalize it to upweight activation for rarely seen scenario and downweight activation for commonly seen ones. We introduce the concept from MDL principle and proposed to consider neural network training process as a model selection problem. We compute the optimal universal code length by normalized maximum likelihood in an incremental fashion, and showed this implementation can be easily incorporated with established methods like batch normalization and layer normalization. In addition, we proposed saliency normalization, which can introduce top-down attention and data prior to facilitate representation learning. Fundamentally, we implemented with an incremental update of normalized maximum likelihood, constraining the implicit space to have a low model complexity and short universal code length.

Preliminary results offered a proof of concept to the proposed method. Given the limited experiments at the current state, our approach empirically outperforms existing normalization methods its advantage in the imbalanced or limited data scenario as hypothesized. Next steps of this research include experiments with variants of the regularity normalization (SN, RLN, RBN etc.), as well as the inclusion of top-down attention given by data prior (such as feature extracted from signal processing, or task-dependent information). In concept, regularity-based normalization can also be considered as an unsupervised attention mechanism imposed on the input data. As the next step, we are currently exploring this method to convolutional and recurrent neural networks, and applying to popular state-of-the-art neural network architectures in multiple modalities of datasets, as well as the reinforcement learning setting where the rewards can be very sparse and non-uniform.

Table 1: Test errors of the imbalanced permutation-invariant MNIST 784-1000-1000-10 task

| | "Balanced" | "Rare minority" | | | "Highly imbalanced" | | | | "Dominant oligarchy" | |
|---|---|---|---|---|---|---|---|---|---|---|
| | $n=0$ | $n=1$ | $n=2$ | $n=3$ | $n=4$ | $n=5$ | $n=6$ | $n=7$ | $n=8$ | $n=9$ |
| baseline | $4.80 \pm 0.34$ | $14.48 \pm 0.63$ | $23.74 \pm 0.63$ | $32.80 \pm 0.48$ | $42.01 \pm 1.01$ | $51.99 \pm 0.71$ | $60.86 \pm 0.42$ | $70.81 \pm 0.90$ | $80.67 \pm 0.81$ | $90.12 \pm 0.56$ |
| BN | $\mathbf{2.77 \pm 0.12}$ | $12.54 \pm 0.68$ | $21.77 \pm 0.57$ | $30.75 \pm 0.68$ | $40.67 \pm 1.01$ | $49.96 \pm 1.02$ | $59.08 \pm 1.56$ | $67.25 \pm 1.21$ | $76.55 \pm 3.15$ | $80.54 \pm 5.31$ |
| LN | $3.09 \pm 0.25$ | $8.78 \pm 1.89$ | $14.22 \pm 1.45$ | $20.62 \pm 3.26$ | $26.87 \pm 2.16$ | $34.23 \pm 4.64$ | $36.87 \pm 1.43$ | $41.73 \pm 6.12$ | $\mathbf{41.20 \pm 2.52}$ | $\mathbf{41.26 \pm 2.90}$ |
| WN | $4.96 \pm 0.26$ | $14.51 \pm 0.98$ | $23.72 \pm 0.87$ | $32.99 \pm 0.62$ | $41.95 \pm 1.03$ | $52.10 \pm 0.67$ | $60.97 \pm 0.40$ | $70.87 \pm 0.88$ | $80.76 \pm 0.80$ | $90.12 \pm 0.56$ |
| RN | $4.91 \pm 0.87$ | $8.61 \pm 1.93$ | $14.61 \pm 1.29$ | $19.49 \pm 1.01$ | $\mathbf{23.35 \pm 2.74}$ | $33.84 \pm 3.77$ | $41.47 \pm 4.28$ | $60.46 \pm 6.45$ | $81.96 \pm 1.31$ | $90.11 \pm 0.54$ |
| RLN | $5.01 \pm 0.65$ | $9.47 \pm 2.70$ | $\mathbf{12.32 \pm 1.25}$ | $22.17 \pm 2.11$ | $23.76 \pm 3.48$ | $32.23 \pm 3.70$ | $43.06 \pm 7.95$ | $57.30 \pm 14.16$ | $88.36 \pm 3.97$ | $89.55 \pm 0.71$ |
| LN+RN | $4.59 \pm 0.65$ | $\mathbf{8.41 \pm 2.59}$ | $12.46 \pm 1.95$ | $\mathbf{17.25 \pm 3.28}$ | $25.65 \pm 4.27$ | $\mathbf{28.71 \pm 4.40}$ | $\mathbf{33.14 \pm 5.58}$ | $\mathbf{36.08 \pm 4.66}$ | $44.54 \pm 3.89$ | $82.29 \pm 9.94$ |
| SN | $7.00 \pm 0.41$ | $12.27 \pm 2.91$ | $16.12 \pm 3.11$ | $24.91 \pm 3.60$ | $31.07 \pm 3.15$ | $41.87 \pm 3.98$ | $52.88 \pm 4.67$ | $68.44 \pm 3.19$ | $83.34 \pm 4.13$ | $82.41 \pm 5.14$ |

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

# A   PRELIMINARY RESULTS

As a proof of concept, we evaluated our approach on MNIST dataset LeCun (1998) and computed the total number of classification errors as a performance metric. As we specifically wish to understand the behavior where the data inputs are non-stationary and highly imbalanced, we created an imbalanced MNIST benchmark to test seven methods: batch normalization (*BN*), layer normalization (*LN*), weight normalization (*WN*), and regularity normalization (*RN*), as well as three variants: saliency normalization (*SN*) with data prior as class distribution, regularity layer normalization (*RLN*) where the implicit space is defined to be layer-specific, and a combined approach where RN is applied after LN (*LN+RN*).

Given the nature of regularity normalization, it should better adapt to the regularity of the data distribution than other methods, tackling the imbalanced data issue by up-weighting the activation of the rare sample features and down-weighting those of the dominant sample features.

To simulate changes in the context (input) distribution, in each epoch we randomly choose $n$ classes out of the ten, and set their sampling probability to be $0.01$ (only 1% of those $n$ classes are used in the training). In this way, the training data may trick the models into preferring to classifying into the dominant classes. For simplicity, we consider the classical 784-1000-1000-10 feedforward neural network with ReLU activation functions for all six normalization methods, as well as the baseline neural network without normalization. As we are looking into the short-term sensitivity of the normalization method on the neural network training, one epoch of trainings are being recorded (all model face the same randomized imbalanced distribution). Training, validation and testing sets are shuffled into 55000, 5000, and 10000 cases. In the testing phase, the data distribution is restored to be balanced, and no models have access to the other testing cases or the data distribution. Stochastic gradient decent is used with learning rate $0.01$ and momentum set to be $0.9$.

When $n = 0$, it means that no classes are downweighted, so we termed it the *"fully balanced"* scenario. When $n = 1$ to $3$, it means that a few cases are extremely rare, so we termed it the *"rare minority"* scenario. When $n = 4$ to $8$, it means that the multi-class distribution are very different, so we termed it the *"highly imbalanced"* scenario. When $n = 9$, it means that there is one or two dominant classes that is 100 times more prevalent than the other classes, so we termed it the *"dominant oligarchy"* scenario. In real life, *rare minority* and *highly imbalanced* scenarios are very common, such as predicting the clinical outcomes of a patient when the therapeutic prognosis data are mostly tested on one gender versus the others, or in reinforcement learning setting where certain or most types of rewards are very sparse.

Table 1 reports the test errors (in %) of eight methods in 10 training conditions. In the balanced scenario, the proposed regularity-based method doesn't show clear advantages over existing methods, but still managed to perform the classification tasks without major deficits. In both the "rare minority" and "highly imbalanced" scenarios, regularity-based methods performs the best in all groups, suggesting that the proposed method successfully constrained the model to allocate learning resources to the "special cases" which are rare and out of normal range, while BN and WN failed to learn it completely (as in the confusion matrices not shown here). In the "dominant oligarchy" scenario, LN performs the best, dwarfing all other normalization methods. However, as in the case of $n = 8$, LN+RN performs considerably well, with performance within error bounds to that of LN, beating other normalization methods by over 30 %. It is noted that LN also managed to capture the features of the rare classes reasonably well in other imbalanced scenarios, comparing to BN, WN and baseline. The hybrid methods RLN and LN+RN both displays excellent performance in the imbalanced scenarios, suggesting that combining regularity-based normalization with other methods is advantageous.

These results are mainly in the short term domain as a proof of concept. Further analysis need to be included to fully understand these behaviors in the long term (the converging performance over 100 epochs). However, the major test accuracy differences in the highly imbalanced scenario (RN over BN/WN/baseline for around 20%) in the short term provides promises in its ability to learn from the extreme regularities.

# B  RELATED WORK

**Normalization.** Batch normalization (BN) performs global normalization along the batch dimension such that for each neuron in a layer, the activation over all the mini-batch training cases follows standard normal distribution, reducing the internal covariate shift (Ioffe & Szegedy, 2015). Similarly, layer normalization (LN) performs global normalization over all the neurons in a layer, and have shown effective stabilizing effect in the hidden state dynamics in recurrent networks (Ba et al., 2016). Weight normalization (WN) applied normalization over the incoming weights, offering computational advantages for reinforcement learning and generative modeling (Salimans & Kingma, 2016). Like BN and LN, we apply the normalization on the activation of the neurons, but as an element-wise reparameterization (over both the layer and batch dimension). In section 3.2, we also proposed the variant methods based on our approach with batch-wise and layer-wise reparameterization, the regularity batch normalization (RBN) and regularity layer normalization (RLN).

**Description length in neural networks.** Hinton & Van Camp (1993) first introduce the description length to quantify neural network simplicity and develop an optimization method to minimize the amount of information required to communicate the weights of the neural network. Zemel & Hinton (1999) considered the neural networks as population codes and used MDL to develop highly redundant population code. They showed that by assuming the hidden units reside in low-dimensional implicit spaces, optimization process can be applied to minimize the model cost under MDL principle. Our proposed method adopt a similar definition of implicit space, but consider the implicit space as data-dependent encoding statistical regularities. Unlike Zemel & Hinton (1999) and Hinton & Van Camp (1993), we consider the description length as a indicator of the data input and assume that the implicit space is constrained when we normalize the activation of each neurons given its statistical regularity. Unlike the implicit approach to compute model cost, we directly compute the minimum description length with optimal universal code obtained in an incremental style.

