# OpenReview forum: "Regularity Normalization: Constraining Implicit Space with Minimum Description Length"
_ICLR.cc/2019/Workshop/LLD — Submitted to LLD 2019_

### Official Review · AnonReviewer2 · 2019-04-06
**Good introduction but unclear theoretical framework and inconclusive experimental evidence**

**Rating:** 1
**Confidence:** 1

**Review:**

This paper provides an adaptation of the minimum description length (MDL) principle in computational neuroscience to propose a new attention mechanism for deep neural networks (DNN), whether in a supervised, unsupervised, or reinforcement learning setting. The authors name this attention mechanism "regularity normalization".

As of now, the manuscript suffers from many imprecisions and inaccuracies, thus hindering its eloquence.

First of all, the authors equate MDL with negative log-likelihood (NLL), but nowhere in the text do they make this connection explicit. Yet, there is a considerable body of literature in minimizing negative log-likelihood for unsupervised learning applications. Maybe i am missing an important distinction between MDL and NLL, but even then, the authors should clarify that distinction.

The definition of the universal code \bar{P}(x) is unclear, and there is no justification of the existence of such universal code.

The following sentence is quite mysterious: "the compressibility of the model will be unattainable for multiple inputs, as the probability distributions are different". It is impossible to know what these "multiple inputs" and "different probability distributions" are.

Many of the claims in the paper lack a proper definition of their terms. An example is: "The normalized maximum likelihood minimizes the worst case regret with the minimax optimal solution". This sentence is very opaque given that the authors haven't explained how they compute regret, what "worst-case" means, and thus what "minimax" corresponds to in their specific setting. The authors should not leave to reader's guesswork the understanding of these technical terms.

The introduction of the paper is well written. The discussion of prior literature, especially in the field of neuroscience, is solid. However, the presentation of the new contribution is too vague to redeem the lack of convincing experimental evidence for its success.

For example, Algorithm 1 (Regularity Normalization) relies heavily on the computation of complexity (COMP in Equation 2), but this equation is not self-contained. On the left-hand side, there is a model class \Theta. On the right-hand side, there are samples x. How does x relate to \Theta? Is the equation integrated over the probability distribution X of the data? None of these questions are properly answered in the text. Therefore, the algorithm is not reproducible. This problem is not limited to Equation (2). Equation (3) is also very difficult to understand, with a dummy variable x_j in the denominator summing up to "0 ... i".

Although this paper exhibits a relatively original and interesting idea for training deep neural networks, i do not recommend it for publication, because it is still in an immature stage. I would recommend the authors to carry out further research, both theoretical and experimental, on regularity normalization, and submit an updated version of this paper in the near future.

---

### Official Review · AnonReviewer1 · 2019-04-13
**Review of "Regularity Normalization: Constraining Implicit Space with Minimum Description Length"**

**Rating:** 4
**Confidence:** 1

**Review:**

The authors present a novel normalization procedure based on weighted maximum likelihood. They extend the idea to make it works for feedforward neural networks.

Globally, the paper is clear and well written, and the contribution interesting.

The algorithm is well-motivated, and the related work is explained clearly.  I like the flexibility of the algorithm, highlighted in section 3.3. However, the step "increment" in Algorithm 1 could be explained more clearly. I think it is worth to include a quick discussion on the complexity of one iteration, and compare it to other regularisation procedure.

The numerical experiments seem promising, and the combination of the regularisation procedure with layernorm gives impressive results when classes are imbalanced.

In conclusion, I recommend acceptance of this paper.

---

### Decision · Program_Chairs · 2019-04-11
**Acceptance Decision**

Reject